# Oligorecurrent Non-Small-Cell Lung Cancer Treated by Chemo-Radiation Followed by Immunotherapy and Intracranial Radiosurgery: A Case Report and Mini Review of Literature

**DOI:** 10.3390/ijms24031892

**Published:** 2023-01-18

**Authors:** Alessio Bruni, Federica Bertolini, Elisa D’Angelo, Giorgia Guaitoli, Jessica Imbrescia, Anna Cappelli, Gabriele Guidi, Alessandro Stefani, Massimo Dominici, Frank Lohr

**Affiliations:** 1Radiation Oncology Unit, Department of Oncology and Hematology, University Hospital of Modena, 41124 Modena, Italy; 2Division of Oncology, Department of Oncology and Hematology, Modena University Hospital, 41124 Modena, Italy; 3PhD Program Clinical and Experimental Medicine, University of Modena and Reggio Emilia, 41124 Modena, Italy; 4Radiation Oncology Unit, Department of Medical and Surgical Sciences for Children & Adults, University of Modena and Reggio Emilia, 41124 Modena, Italy; 5Medical Physics Unit, University Hospital of Modena, 41124 Modena, Italy; 6Division of Thoracic Surgery, Department of Medical and Surgical Sciences, University of Modena and Reggio Emilia, 41124 Modena, Italy; 7Laboratory of Cellular Therapy, Division of Oncology, Department of Medical and Surgical Sciences for Children and Adults, University of Modena and Reggio Emilia, 41124 Modena, Italy

**Keywords:** NSCLC, chemo-radiotherapy, immunotherapy, SABR

## Abstract

Locally advanced non-small-cell lung cancer still represents a “grey zone” in terms of the best treatment choice and optimal clinical outcomes. Indeed, most patients may be suitable to receive different treatments with similar outcomes such as chemo-radiotherapy (CHT-RT) followed by immunotherapy (IO) or surgery followed by adjuvant local/systemic therapies. We report a clinical case of a patient submitted to primary thoracic surgery who developed a mediastinal nodal recurrence successfully treated by CHT-RT-IO. Subsequently, a single brain lesion was found to have been successfully treated by single fraction stereotactic ablative radiotherapy. The patient is still on follow-up and she is free from disease having a good quality of life. In this report, we also perform a mini review about the role of CHT-RT followed by IO in treating loco-regional relapse after surgery. The role of SABR after IO is also evaluated, finding that it is safe and well tolerated. More robust and larger clinical data are needed in this particular setting to better define the role of the combination of systemic and local treatments in the management of intrathoracic and intracranial relapse for patients already submitted to CHT-RT followed by immunotherapy.

## 1. Introduction

Recently, the results of a phase III randomized Pacific trial on patients with unresectable locally advanced non-small-cell lung cancer (NSCLC) submitted to chemo-radiotherapy (CHT-RT) followed by consolidative immunotherapy (IO) confirmed a significant benefit in terms of long-term clinical outcomes, as reported in the update analysis from Spiegel et al. [1]. Furthermore, several observational studies involving large populations obtained similar results in real-world settings underlying the crucial role of this new multimodal approach in stage III NSCLC patients who have not progressed after radical chemotherapy (CHT) and radiotherapy (RT) [2]. Five-year survival outcomes from the Pacific trial showed a median overall survival (OS) and progression free survival (PFS) equal to 47.5 and 16.9 months, respectively, but, unfortunately, 6.3% of the patients developed brain metastasis during or after the consolidation phase with durvalumab [3].

Meanwhile, stage III patients treated by upfront surgery have a risk of developing loco-regional relapse, especially if they are not submitted to adjuvant RT, as recently reported in the lung ART trial [4]. Unfortunately, almost 30% of patients usually experience loco-regional or distant recurrence during their lifetime, thus heavily reducing their expectancy and quality of life. However, selected patients experiencing loco-regional relapse after surgery seem to have favorable clinical outcomes if treated with a multimodal approach such as CHT and RT. More recently, case reports as well as small retrospective series about consolidative durvalumab after CHT-RT were published in this setting [5].

However, the open question is still the same: what therapy (local, systemic, or both) should be considered the optimal choice in patients with (oligo)progression after first-line treatment with CHT-RT and maintenance IO [6]?

The aim of our report is to describe a clinical case of a patient affected by nodal mediastinal recurrence after lobectomy submitted to CHT-RT followed by durvalumab who developed a single brain metastasis (almost at the end of consolidative immunotherapy) treated by stereotactic ablative radiotherapy (SABR). We also conducted a mini review about salvage treatment with or without IO after local recurrence in resected NSCLC patients.

## 2. Case Presentation

A 50-year-old female former smoker (30 pack/year, quit almost 1 year before CHT-RT treatment) was referred to our Institute in July 2019. She had good performance status and no significant comorbidities.

Her oncological history begun in 2017 when she was submitted to upper left lobectomy and homolateral hilar nodal sampling (May 2017) with pathologic report positive for solid pleomorphic neoplasm with spindle and giant cells. The final pathological stage was pT2a R0 pN0 (0/3) cM0. Adjuvant CHT with cisplatin and vinorelbine was administered and concluded on September 2017. During regular follow-up, the patient had a parenchymal relapse on the right lung so that in February 2018 she underwent double atypical right lung resection (dorsal segment of the upper lobe and lateral segment of the lower lobe). The pathological specimen was positive for invasive adenocarcinoma with papillary and solid growth, final stage pT2 pNx. The patient was followed up until July 2019 when an 18 fluor-deoxy-glucose PET-CT scan showed mediastinal recurrence in the right lower paratracheal area (station 4R), close to the bronchial bifurcation. Endobronchial ultrasound (EBUS) confirmed the diagnosis of nodal lung adenocarcinoma recurrence with Programmed Death-Ligand1 (PDL-1) expression 60% and wild type for oncogene alterations. During the multidisciplinary discussion, concomitant CHT-RT with radical intent was proposed. In September 2019 she underwent two cycles of CHT with cisplatin 80 mg/m^2^ and pemetrexed 500 mg concomitant with thoracic RT. The total dose delivered to the right hilar, paratracheal (4R, 11R), and subcarinal (station 10) nodes was 60 Gray in 30 fractions using Tomotherapy^@^. (Figure 1). All treatments delivered were well tolerated with patient referring G2-G3 dysphagia and esophageal mucositis so the patient was fed by parenteral nutrition for a week. Her quality of life during this first phase of treatment mildly decreased for a while, but the patient fully recovered after two weeks from the end of RT treatment.

Total body computed tomography (CT) scan was performed almost 30 days after the last session of RT showing partial response of the right hilar nodes and no further disease metastasis. Thus, the patient underwent consolidative IO with durvalumab (first dose in January 2020). In October 2020, a restaging CT scan showed stable disease of the thoracic lesion, while a single rounded frontal cortical contrast enhanced small lesion at the vertex (diameter of about 8 mm) appeared. The same cerebral metastasis was confirmed by contrast enhanced brain magnetic resonance (MR) with no other intracranial lesions. In the meanwhile, she received the last planned dose of durvalumab (22 cycles). The case was then referred to our lung multidisciplinary meeting team and it was recommended to proceed with single session SABR (December 2020) to the left frontal brain lesion for a total dose of 20 Gray (Gy). Image guided with intensity modulated RT was performed. (Figure 2). Both treatments (systemic and SABR) were well tolerated and no side effects were reported. Her quality of life was always well maintained.

Restaging CT scan (April 2021) showed intrathoracic (right hilar lymph nodes) stable disease; no other visceral or bone lesions were detected. Brain MRI showed partial response of the metastatic intracranial cortico–subcortical lesion treated by SABR. In January 2022 a restaging CT scan was performed showing the appearance of solid tissue with axial extension of about 23 × 13 mm at the emergence of the right lower lobe bronchus, suspected for disease recurrence. An 18FDG PET scan did not reveal significant intrathoracic uptake so the patient is still (September 2022) on regular follow-up without receiving second-line systemic therapy. The study was conducted in accordance with the Declaration of Helsinki. The patient gave written informed consent to report her clinical information.

## 3. Discussion

Nodal relapse from resected NSCLC is commonly reported in daily clinical practice, especially in patients with pathological stage III N2 NSCLC not submitted to adjuvant loco-regional treatment. The gold standard treatment is still not completely clarified in this setting due to the multiple therapeutic options available such as local (RT) or systemic (CHT, IO) treatments that may be prescribed alone or within a multimodal approach [7,8]. The recent publication of the Lung ART trial showed that adjuvant RT should not be performed due to the lack of statistically significant advantages in terms of overall survival (3-year overall survival 69% in the control group and 67% in the PORT one) [4]. However, analyzing disease free survival, the authors reported that 106 of 296 patients in which at least an event occurred during follow-up, had mediastinal relapse (36 (25%) of 144 patients in the PORT group and 70 (46%) of 152 patients in the control group) highlighting a clinical difference in terms of local relapse in favor of the PORT group. On the other side, IMPOWER-010 trial recently showed a significant advantage in patients submitted to adjuvant atezolizumab after complete surgery for stage I-III NSCLC [9]. In particular, patients with completely resected (negative margins) stage IB-IIIA with mediastinal lymph node dissection at specified levels, able to receive cisplatin-based chemotherapy, were randomized to consolidative atezolizumab or observation. The risk of recurrence in the experimental arm was reduced by 34% in the stage II–IIIA population whose tumors expressed PD-L1 >1% and by 21% in all patients with stage II–IIIA. Due to these findings, it is expected that adjuvant RT may be more often avoided, particularly in selected patients with a low risk of loco-regional relapse. Furthermore, as reported in the lung ART trial, even more patients will experience nodal recurrence during their lifetime, thus having the chance of being submitted to the multimodal approach comprehensive of systemic (CHT and/or IO) and local treatment (RT). Recently, Borghetti et al. conducted a multicentric retrospective study on 24 patients submitted to CHT-RT followed by consolidative IO using durvalumab [10]. The median PFS was 15 months; the 12-, 18-, and 24- month PFS rates were 68.7%, 45.8%, and 34.3%, respectively, confirming interesting results in terms of clinical outcomes. Furthermore, the safety profile was optimal with Grade 3 or 4 adverse events occurring in only two patients. The most common side effect was esophagitis (58.3%) followed by hematological toxicity (20.8%). All toxicities resumed with appropriate supportive therapies within some weeks from the end of RT. These findings seem to confirm the crucial role of intensifying a systemic approach in patients with loco-regional relapse after radical surgery for early-stage NSCLC. Indeed, as already reported by Terada et al. [11], the most common pattern of failure in this setting was characterized by a metastatic extra thoracic spread confirming the importance of adding new drugs such as IO in the management of relapsed NSCLC. This rationale is already described in a very similar setting represented by unresectable stage III-N2 NSCLC treated by concurrent CHT-RT followed by durvalumab in which, at a median follow-up of 25.2 months, OS was improved with durvalumab obtaining 12- and 24-month OS rates of 84.5% and 68.3%, respectively [12]. Similar results were recently confirmed in a multicentric retrospective series involving more than 180 patients affected by unresectable stage III NSCLC. In this trial, the addition of consolidative durvalumab may allow very interesting results to be achieved in terms of clinical outcomes with 12, 18, and 24-month PFS rates of 83.5%, 65.5%, and 53.1%, respectively and 12, 18, and 24-month OS rates of 97.2%, 87.9%, and 79.3%, respectively [2]. The authors analyzed the pattern of failures showing that 55 of 187 patients (35.5%) relapsed locally or systemically, 20.6% had loco-regional progression, 46 (29.7%) developed systemic metastases, while 23 patients (14.8%) had both local and systemic recurrence, and, finally, 23 (14.8%) had systemic spread alone. Furthermore, 30 patients (54.5%) received systemic treatments (most of all CHT) at the time of progression, while nine (16.3%) underwent metastasis-directed SABR and the remaining 16 patients (29%) were referred to palliative care. For the abovementioned reasons, we can affirm that selected patients with nodal mediastinal recurrence after surgery should be considered as well as those patients who were candidates for the PACIFIC regimen. Similar findings were found in a recent review published by Brooks et al. in which salvage treatments for loco-regional relapse after SABR for early-stage NSCLC were reported [13]. In this review, patients experiencing regional recurrence (whether isolated or in combination with parenchymal lesion) seemed to have a very similar OS to the one obtained in patients with newly diagnosed unresectable stage III NSCLC submitted to CHT-RT followed by IO. Indeed, the same author had previously published the results of a prospective study on more than 900 patients submitted to primary SABR and then experiencing a regional relapse. In this study, a multimodal approach concerning systemic and local therapy showed a significant benefit in terms of clinical outcomes, particularly for patients with isolated regional relapse. In these patients, bimodality treatment with nodal irradiation and systemic therapy was preferred with most patients receiving a platinum-based regimen together with conventional RT to the involved nodes for a total dose of 60 to 70 Gy in 2 Gy fractions [14]. Among the patients with isolated regional relapse, grade 3 or more effects occurred in 10 of 26 patients who had CHT-RT (38.5%; esophagitis, fatigue, and hematologic effects), 1 of 8 who had conventional RT (12.5%; dyspnea), and 4 of 12 who had systemic therapy (33.3%; most common was fatigue), also confirming the optimal safety profile in this unfavorable setting. Furthermore, the OS for patients with isolated regional recurrence, even if treated with salvage multimodal treatment, was poorer than that for patients with isolated parenchymal relapse but similar to that for patients with unresectable stage III disease. The last crucial point of our analysis is the role of metastasis-directed therapy such as SABR when oligoprogression occurs during or after the maintenance systemic therapy. Recently, results of a prospective observational study were published showing the importance of the combination of local treatment (SABR) and systemic IO in stage IV NSCLC and melanoma patients. In this study, Chicas-Sett and colleagues [15] delivered immunostimulating SABR (35 Gy in 5 fractions or 24 Gy in 3) to up to five measurable lesions (nodal, visceral, bone, or brain) in association with pembrolizumab or nivolumab. Particularly at a median follow-up of 32.8 months, the objective response rate was 42% with a median PFS and OS of 14.2 and 37.4 months, respectively. Furthermore, approximately 50% of NSCLC patients developed BM during their disease, thus having a very short life expectancy. Recently, Scoccianti et al. [16] published the results of an Italian multicentric study involving 100 patients with 163 brain metastases treated using SABR (alone or concomitantly with IO). In particular, the combination of SABR plus IO resulted in a 1-year intracranial local PFS of 83.9% that could be considered satisfying in stage IV NSCLC. Moreover, patients treated with SABR within 7 days from the administration of IO had a longer survival when compared to patients with the interval between SABR and IO > 7 days (propensity score-adjusted *p* = 0.007), once more demonstrating that the concurrent administration (even if the definition is extremely heterogeneous in the existing literature), should be preferred in this setting. If evidence regarding stage IV increases, the management of isolated distant relapse during durvalumab consolidation in stage III NSCLC is still an unmet need. According to different real-world series, 20–30% of patients experience distant relapse during durvalumab treatment, with the brain being one of the most common sites of relapse [17,18,19]. To the best of our knowledge, there is no evidence about the best therapeutic management in this setting, or about a possible durvalumab continuation and integration with loco-regional treatments. Indeed, if this approach appears safe, it is not possible to draw any conclusion about its effectiveness on patients’ outcomes. In the absence of answers from the international literature, many factors should probably be taken into account, including the site of the relapse, timing of the relapse during IO consolidation, and the alternative first-line options.

## 4. Conclusions

Loco-regional relapse is still a great concern in the management of NSCLC with no clear standard of care. Recently, IO has demonstrated the ability of hardly slowing the clinical progression of stage III and IV NSCLC. Our study confirms the importance of a multimodal approach comprehensive of CHT-RT followed also by consolidative IO in patients affected by loco-regional relapse after surgery. The combination of CHT-RT plus IO seems to be safe and tolerable with interesting results in terms of clinical outcomes, even if it should be more deeply investigated in more robust prospective clinical trials.

## Figures and Tables

**Figure 1 ijms-24-01892-f001:**
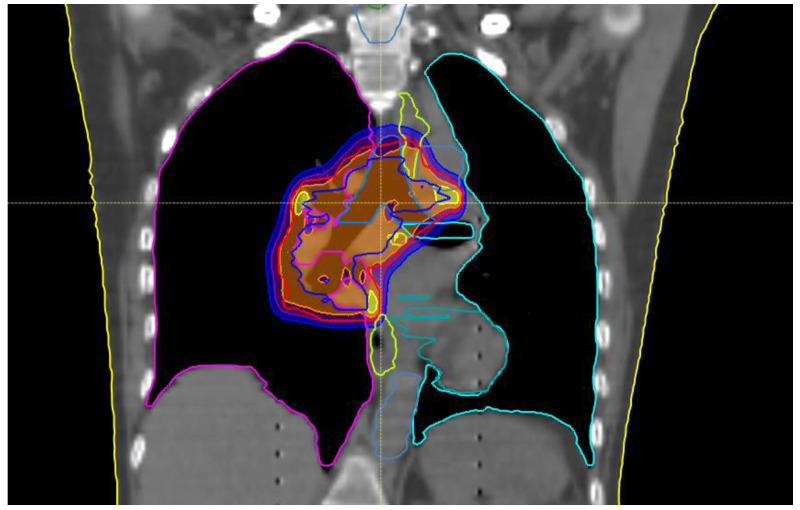
Treatment plan for mediastinal relapse.

**Figure 2 ijms-24-01892-f002:**
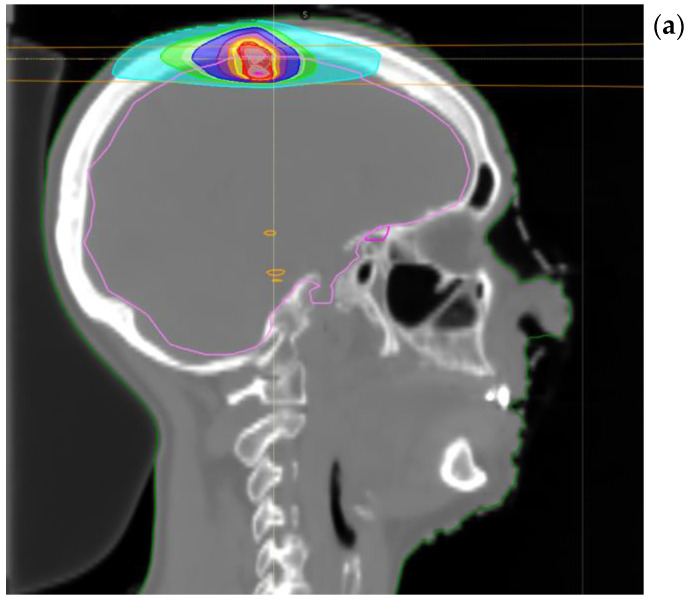
Treatment plan for single session brain SABR (**a**) sagittal and (**b**) coronal reconstruction.

## Data Availability

Not applicable.

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
