# Peer review of "Oligorecurrent Non-Small-Cell Lung Cancer Treated by Chemo-Radiation Followed by Immunotherapy and Intracranial Radiosurgery: A Case Report and Mini Review of Literature"

_ijms, 2023, doi:10.3390/ijms24031892_

Round 1

Reviewer 1 Report

The paper describes a case of NSLC treated with upper left lobectomy and omolateral hilar nodal sampling (stage pT2a R0, pN0) followed by adjuvant chemotherapy. in the following years the patient presented local relapses treated with chemo-radiotherapy and consolidative immunotherapy with durvalumab. In addition, for a single brain metastasis the patient underwent stereotactic ablative radiotherapy. Actually the disease is stable. The case report is well-descrbed and of interest because is difficult, in this setting of patients, find the best therapy. these experience evidences the possible important role of combination therapy with durvalumab and loco-regional treatments.

Some little remarks:

- line 21, please specify the Pacific trial

- Please specify the quality of life of the patient during the treatments

- Did the patient well-tolerated SABR?

- In your opinion is possible to administer this type of tratment to older patients?

Author Response

Dear Reviewer,

thank you very much for your comments and suggestions. I try to answer to your response in a point-by-point way:

- At line 21 I specified the design of the Pacific trial as requested.

- At line 71-74 and 81 I added the tolerance to the treatments delivered and the quality of life of the patient as requested.

- At line 81 I highlighted how the safety of SABR was great, no side effects were reported during the following follow up.

Finally a little comment to your last question. In my opinion the treatment is substantially well tolerated and I think it's possible to offer it also to selected older patients, particularly if they are fit for chemotherapy. I may suggest to strictly follow up this patients, in particular if they are frailer or older also using multimodal approach such as the active involvment of nutritionists, care givers, geriatric speacialists.

Reviewer 2 Report

Review report

This article titled “Oligorecurrent non small cell lung cancer treated by chemo-radiation followed by immunotherapy and intracranial radiosurgery:a case report and mini review of literature

The case report is well explained with supporting evidence.

The introduction was well written and explained in detail.

2. Case Presentation: after line number 57, need to include the following reference,

Fuloria S, Subramaniyan V, Karupiah S, Kumari U, Sathasivam K, Meenakshi DU, Wu YS, Sekar M, Chitranshi N, Malviya R, Sudhakar K, Bajaj S, Fuloria NK. Comprehensive Review of Methodology to Detect Reactive Oxygen Species (ROS) in Mammalian Species and Establish Its Relationship with Antioxidants and Cancer. Antioxidants (Basel). 2021;10(1):128. doi: 10.3390/antiox10010128. PMID: 33477494; PMCID: PMC7831054.

3. Discussion: after line number 99, need to include the following reference,

Subramaniyan V, Fuloria S, Gupta G, Kumar DH, Sekar M, Sathasivam KV, Sudhakar K, Alharbi KS, Al-Malki WH, Afzal O, Kazmi I, Al-Abbasi FA, Altamimi ASA, Fuloria NK. A review on epidermal growth factor receptor's role in breast and non-small cell lung cancer. Chem Biol Interact. 2022 Jan 5;351:109735. doi: 10.1016/j.cbi.2021.109735. Epub 2021. PMID: 34742684.

Discussion section is well explained and correlated with previous studies.

The conclusion is well defined.

Report

The entire case report is well written. This paper can be published after minor revisions.

Author Response

Dear Reviewer,

thank you very much for your review and for your valuable suggestions.

As suggested, I added the abovementioned studies in the reference , both at line 107-108 in the discussion graph that in my opinion is the best place instead of being inserted during the case presentation.

thank you again

Alessio Bruni